# Metal-organic frameworks as kinetic modulators for branched selectivity in hydroformylation

Gerald Bauer [1], Daniele Ongari [2], Davide Tiana [3], Patrick Gäumann [1], Thomas Rohrbach[1], Gerard Pareras [3], Mohamed Tarik [4], Berend Smit [2] & Marco Ranocchiari [1✉]

Finding heterogeneous catalysts that are superior to homogeneous ones for selective catalytic transformations is a major challenge in catalysis. Here, we show how micropores in metal-organic frameworks (MOFs) push homogeneous catalytic reactions into kinetic regimes inaccessible under standard conditions. Such property allows branched selectivity up to 90% in the Co-catalysed hydroformylation of olefins without directing groups, not achievable with existing catalysts. This finding has a big potential in the production of aldehydes for the fine chemical industry. Monte Carlo and density functional theory simulations combined with kinetic models show that the micropores of MOFs with UMCM-1 and MOF-74 topologies increase the olefins density beyond neat conditions while partially preventing the adsorption of syngas leading to high branched selectivity. The easy experimental protocol and the chemical and structural flexibility of MOFs will attract the interest of the fine chemical industries towards the design of heterogeneous processes with exceptional selectivity.

[1] Laboratory for Catalysis and Sustainable Chemistry, Paul Scherrer Institute, 5232 Villigen PSI, Switzerland. [2] Laboratory of Molecular Simulation (LSMO), Institut des Sciences et Ingénierie Chimiques, Valais, Ecole Polytechnique Fédérale de Lausanne (EPFL), Rue de l'Industrie 17, CH-1951 Sion, Switzerland. [3] School of Chemistry, University College Cork, College Road, Cork, Ireland. [4] Laboratory for Bioenergy and Catalysis, Paul Scherrer Institute, 5232 Villigen PSI, Switzerland. ✉email: marco.ranocchiari@psi.ch

The fine chemical industry is dominated by homogeneous molecular catalysts when high selectivity is desired, such as in regioselective and in stereoselective transformations. The main strategy to introduce heterogeneous catalysts to the fine chemical industry has been to immobilize homogeneous molecular catalysts on mesoporous supports and insoluble nanoparticles or polymers to overcome diffusion limitations and to accommodate large molecular active sites. Although some immobilized catalysts show promising catalytic activity, this strategy is still not applied in the chemical industry since "heterogeneization" is simply not enough[1]. There is a need to demonstrate how the use of heterogeneous catalysts can promote selectivity that are challenging or even impossible to be obtained with existing catalytic systems. This can be done only if the chemical properties of heterogeneous catalysts can go beyond easier separation and recycling.

Since the discovery of metal-organic frameworks (MOFs), many researchers have been looking for catalytic applications with unique performance[2–5]. The chemical flexibility, tuneable pore size and chemical and structural stability of MOFs showed how they can be used to design active sites at the molecular level to direct selectivity and performance of reactions[6–12]. In recent years, promising catalytic applications that use MOFs as precursors for novel materials[13] as well as model systems to understand heterogeneous catalysis processes have been described[14,15]. After several decades, the field of catalysis by MOFs is still in its infancy since most of the examples are proof of concepts and do not offer attractive advantages to existing catalysts[1,16]. MOFs are widely known for their ability to selectively adsorb different molecules depending on their structure. This is a unique feature available only to microporous materials[17–19].

The hydroformylation of olefins—or oxo synthesis (Fig. 1)—is one of the most important reactions catalysed by homogeneous catalysts to obtain aldehydes from olefins in the presence of syngas[20]. The atom economic process yields linear aldehydes and branched ones. The linear isomers are key intermediates for the detergent and polymer industry and are formed with Rh catalysts, which are generally more selective than Co ones[21]. Branched aldehydes are a powerful tool for the fine chemical industry with potential applications in the formation of enantio-enriched products. Rh catalysts with bidentate ligands dominate the scene, especially with substrates with directing groups[22]. At present, the branched-selective hydroformylation of olefins without directing groups is still challenging and can be achieved only by complex Rh catalysts with a selectivity for 2-methylhexanale from 1-hexene up to 75%[23,24] and up to 86% for 2-methylbutanale from 1-butene[25]. Supramolecular chemistry has also been used to tune regioselectivity in Rh-catalysed hydroformylation[26–28]. The chemistry of Co-catalysed "branched-selective" hydroformylation is rare and yields at best moderate selectivity of acetal-protected products[29].

In this contribution, we demonstrate how adsorption properties can be exploited in catalysis to get otherwise inaccessible kinetics under standard conditions. We show that the micropores of MOFs push the Co-catalysed hydroformylation of olefins without directing groups to kinetic regimes that favour high branched selectivity.

## Results and discussion

**The selectivity limit of homogeneous hydroformylation.** Several catalytic conditions were screened, aiming to maximize the yield of the branched product with 1-hexene as substrate and $Co_2(CO)_8$ as pre-catalyst to identify the highest branched selectivity that may be obtained in homogeneous catalysis (Supplementary Table 5). Preliminary reactions at different temperatures and pressures showed an optimum temperature at 100 °C—at higher ones, a significant amount of the isomerization of the olefin was observed—and 30 bar syngas due to the lower branched selectivity at higher pressures. The reaction mixture without MOF showed a conversion of 40% with branched to linear ratio (B/L) of 49:51. The only way to exceed the 50% selectivity threshold was to reduce the pressure to 19 bar, which resulted in 15% conversion with 66% selectivity towards the branched products, 2-methylhexanal (**1**) and 2-ethylpentanal (**2**) (Fig. 1a), which were formed in a 3:1 ratio. Increasing the catalyst loading to 0.47 $mol_{Co}$%, 1.19 $mol_{Co}$% and 2.38 $mol_{Co}$% led to 61%, >99% and >99% conversion, respectively with a B/L of 1:1 (Supplementary Table 5). Only a narrow range of experimental conditions led to moderate branched selectivity at low conversion rate, which is consistent with kinetic studies[30,31]. In homogeneous catalysis, the limit to achieve high branched selectivity is to work in neat 1-hexene at 100 °C and 19 bar syngas pressure. As demonstrated below, we can go beyond this limit and achieve much higher branched selectivity by adding MOFs to the reaction mixture.

**The addition of MOFs enhances branched selectivity.** Our group has previously shown that MOFs with UMCM-1 topology can fully adsorb chiral Rh complexes within the pores of the frameworks leading to an increased performance in the asymmetric hydrogenation of olefins[32]. The addition of UMCM-1 and UMCM-1-NH$_2$ (fully or partially functionalized, Fig. 2a) in the hydroformylation of 1-hexene with $Co_2(CO)_8$ (0.23 $mol_{Co}$%) was tested (Table 1). All screening was performed at the conditions that gave around 40% conversion and 50% selectivity in the pure homogeneous system (Table 1, entry 1) at 100 °C, 30 bar syngas with neat 1-hexene. MOFs with UMCM-1 topology (Fig. 2a) gave 60%, 76% and 76% branched selectivity, respectively, for UMCM-1 (Table 1), MixUMCM-1-NH$_2$ (28%) (Table 1) and UMCM-1-NH$_2$ (Table 1), all at around 30% conversion.

We investigated further the effect of MixUMCM-1-NH$_2$ (28%) additive. A screening of the amount of MOF was performed by keeping constant the Co catalyst concentration at 0.23 $mol_{Co}$% (Supplementary Table 6), which revealed that there is an optimal MOF/Co molar ratio of 0.8. The branched products can be obtained in 75:25 B/L ratio with a **1** and **2** ratio of 3:1 and 36%

**Fig. 1 Co-catalysed hydroformylation. a** General scheme. For 1-hexene R = C$_4$H$_9$, R$_1$ = C$_3$H$_7$. **b** Accepted mechanism for the co-catalysed hydroformylation of olefins.

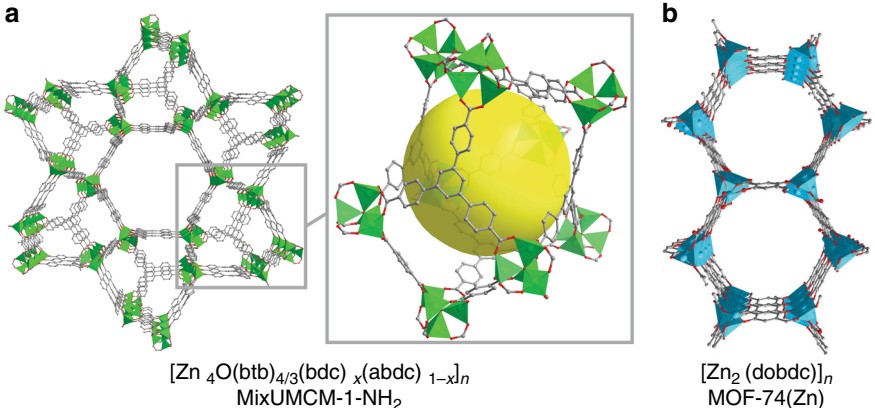

$[Zn_4O(btb)_{4/3}(bdc)_x(abdc)_{1-x}]_n$
MixUMCM-1-NH$_2$

$[Zn_2(dobdc)]_n$
MOF-74(Zn)

**Fig. 2 Structures and molecular formulas of the MOFs used in hydroformylation. a** MixUMCM-1-NH$_2$. **b** MOF-74(Zn). bdc = 1,4-benzenedicarboxylate; abdc = 2-amino-1,4-benzenedicarboxylate, btb = 4,4',4'',-benzene-1,3,5-triyl-trisbenzoate and dobdc = 2,5-dioxido-1,4-benzenedicarboxylate. Hydrogen and nitrogen atoms are omitted for clarity.

**Table 1 Influence of MOFs on the selectivity and reactivity of the Co-catalysed hydroformylation[a].**

| Entry | Additive | Conversion (%)[b] | B/L[c] |
|---|---|---|---|
| 1 | None | 40 | 49:51 |
| 2 | UMCM-1 | 32 | 60:40 |
| 3 | MixUMCM-1-NH$_2$ (28%) | 24 | 76:24 |
| 4 | UMCM-1-NH$_2$ | 20 | 76:24 |
| 5 | MixUMCM-1-NH$_2$ (28%)[d] | 36 | 75:25 |
| 6 | MOF-74(Zn)[e] | 25 | 85:15 |

[a]Co$_2$(CO)$_8$ (0.8 mg, 2.3 μmol) were dissolved in 1-hexene (250 μL, 2.0 mmol) and the MOF was added (mol$_{MOF}$/mol$_{Co}$ = 1.7); the mixture was brought to 30 bar and 100 °C for 17 h.
[b]The final reaction mixture contained 1-hexene, **1**, **2**, n-heptanale and unknown compounds (~2%) as detected by GC-FID and GC-MS.
[c]**1**:**2** ratio = 3:1.
[d]mol$_{MOF}$/mol$_{Co}$ = 0.8.
[e]mol$_{MOF}$/mol$_{Co}$ = 20.

conversion (Table 1, entry 5). This is a remarkable improvement showing that good branched selectivity can be achieved while maintaining the conversion levels of homogeneous catalysis. Inductively coupled plasma mass spectrometry (ICP-MS) showed that MixUMCM-1-NH$_2$ (28%) adsorbed 73% of the Co after reaction under the conditions in Table 1, entry 5 (Supplementary Table 3). The enhanced branched selectivity obtained by adding MixUMCM-1-NH$_2$ (28%) compared to the homogeneous reaction is observed at all pressures ranging from 19 to 78 bar. The higher the pressure the lower the overall branched selectivity, though (Supplementary Table 6). One can increase the Co adsorption (up to 85%) with pressures up to 72 bar, whereas the cobalt adsorbed in the MOF decreases to 67% at 94 bar (Supplementary Table 3). A series of blank experiments with additives such as the linkers in the MOFs and different Zn sources were done to rule out that leached species and defects enhance branched selectivity (Supplementary Table 7).

We tested different MOF topologies that have been synthesized in our labs aimed at understanding the role of the MOF environment in such a change in selectivity. MOF-74(Zn) (Fig. 2b) is superior to MixUMCM-1-NH$_2$ (28%) giving 25% conversion and 85% selectivity (Table 1 entry 6) while absorbing 60% of the Co. Both MixUMCM-1-NH$_2$ (28%) and MOF-74(Zn) retained crystallinity after catalysis as shown by powder X-ray diffraction (Supplementary Fig. 4). The BET number of MixUMCM-1-NH$_2$

(28%) changed from 2870 to 2980 m$^2$/g after catalysis (Supplementary Figs. 5 and 7), whereas the one of MOF-74(Zn) decreased from 1000 to 150 m$^2$/g (Supplementary Figs. 6 and 8). The lower surface area in MOF-74 after catalysis is caused by the adsorption of the catalyst and of organic products such as aldol compounds (Supplementary Fig. 10), which could not be removed by extensive washing, as also evidenced by pore size distributions before and after catalysis and by the recycling experiments below.

We attempted incipient wetness impregnation of the Co$_2$(CO)$_8$ pre-catalyst in dichloromethane at MOF/Co molar ratio of 0.7 and 0.3 with UMCM-1-NH$_2$ and MOF-74(Zn), respectively, to check whether the pre-adsorbed metal would lead to higher selectivity. Once 1-hexene was added to the impregnated MOF, a strong coloration of the homogeneous solution to dark brown was observed in all experiments, indicating that the Co complex was preferentially in solution. Catalytic results under such conditions showed up to 41% conversion and 68% branched selectivity with 0.46 mol$_{Co}$% for UMCM-1-NH$_2$ and up to 63% conversion and 68% branched selectivity with a 0.6 mol$_{Co}$% loading in the case of MOF-74(Zn) (Supplementary Table 11), showing that higher selectivity can be obtained by adding the MOF to the reaction mixture rather than with pre-formed impregnated pre-catalyst. This is attributed to the relatively low chemical stability of Co$_2$(CO)$_8$[33] and suggests that it is an intermediate of the catalytic cycle that preferentially adsorbs within MOFs rather than the pre-catalyst.

After the first catalytic runs in Table 1, entries 5 and 6, we extensively washed the MOFs and tested their recyclability for a second catalytic run. MixUMCM-1-NH$_2$ (28%) showed 5% conversion and 64% branched selectivity, whereas MOF-74(Zn) exhibited 68% selectivity at 35% conversion (Supplementary Table 13). Branched selectivity decreased in both recycling experiments compared to the first run, but it was still higher than in the homogeneous reaction. Higher conversion could be obtained with MOF-74(Zn). All recycling results suggest that there are trace compounds that are hard to wash out and may modify the adsorption properties of the MOFs causing lower selectivity as suggested also by BET after catalysis (see above). The recycling of impregnated Co@MOF catalysts with minimal washing—to keep the Co inside the pores—resulted in inactive catalysts (Supplementary Tables 2 and 12).

MOF-74 with different metals (Supplementary Fig. 2 and Supplementary Table 1) such as Mg gave 77% branched products at 8% conversion, whereas MOF-74(Co) and MOF-74(Ni) both yielded around 55% branched aldehydes at 90% and 44% conversion, respectively. While MOF-74(Mg) is of little use since

it lowers dramatically conversion and adsorbs a little Co amount (~20%), MOF-74(Co) and MOF-74(Ni) are reactive towards syngas and did not increase branched selectivity. MIL-101-NH$_2$(Al), MIL-101(Cr) (Supplementary Fig. 3 and Supplementary Table 1) and zeolite-Y were also tested, but they gave almost no conversion and killed the catalytic activity (Supplementary Table 8).

There are two possible reasons why this selectivity enhancement is observed by adding certain MOFs to homogeneous catalysis: (1) either there is a bond between the Co complex and the MOF leading to a coordinative interaction, which consequently changes the reactivity of the active site, or (2) the micropores of the MOFs alter the kinetics and energetics along the reaction pathway.

**Electronic interactions are unlikely.** We performed a set of computational and experimental studies to investigate what could be the cause of the selectivity change. Co$_2$(CO)$_8$ forms the active pre-catalyst HCo(CO)$_4$ in the presence of syngas (Fig. 1, step **A**) and upon decoordination of one carbonyl group the active catalyst HCo(CO)$_3$ (Fig. 1, step **B**) is formed. We modelled interaction energies of both species within the pores of MOF with UMCM-1 and MOF-74(Zn) topologies with density functional level of theory (DFT) (Supplementary Information). The interaction energies between HCo(CO)$_4$ and the MOFs with UMCM-1 (Supplementary Figs. 11 and 12) and MOF-74 (Supplementary Fig. 13) topologies are between +1.2 and −2.53 kcal/mol (Supplementary Table 14) and therefore in the range of van der Waals interactions and not of a coordination bond. When inside the pores of the MOF, the formation of a MOF–Co(H)(CO)$_3$ system can be envisioned, though (Supplementary Table 15). The stabilization energy of unfunctionalized UMCM-1–Co(H)(CO)$_3$ is 3.5 kcal/mol (Supplementary Fig. 14) and therefore negligible to form a coordinative bond. The functionalized UMCM-1-NH$_2$ (Supplementary Fig. 15) and MOF-74(Zn) (Supplementary Fig. 16) stabilize the unsaturated complex HCo(CO)$_3$ with 20–35 kcal/mol stabilization energy. Such energies are significantly lower than the binding energy between CO–Co(H)(CO)$_3$ (−62.7 and −61.6 kcal/mol for the axial and equatorial CO, respectively) and 1-hexene–Co(H)CO$_3$ (−52.8 kcal/mol), which are formed under reaction conditions. The DFT energies above suggest that it is unlikely that a coordinative bond between the unsaturated cobalt complex and the MOF is formed under catalytic conditions. This is supported by analysing the interaction of the pre-catalyst Co$_2$(CO)$_8$ with the MOFs by infrared spectroscopy of impregnated Co@MOFs (Supplementary Fig. 9). The inactivity of MIL-101 and zeolite-Y might be explained by an electronic interaction between the metal open sites in the MOF and the Co catalyst. The detrimental effect of Al/amines and of Cr to the activity in Co-catalysed hydroformylation is described in the literature[34,35].

Further evidence that no electronic interaction is responsible for the selectivity change was provided by preparing MOFs that feature strong coordinative bonds with Co such as phosphine MOFs (P-MOFs)[15,36–38]. Such phosphine solid ligands can coordinate Co as supported by the literature[39] and by our DFT P–Co binding energies (Supplementary Table 15 and Suplementary Fig. 17). MixUMCM-1-PPh$_2$ (29%) was synthesized (Supplementary Fig. 1) and further tested in catalysis. The pre-formed HCo(CO)$_3$(Mix-UMCM-1-PPh$_2$) complex[40] formed 50% branched aldehydes at 9% conversion showing that the P–Co bond does not yield branched selectivity. The addition of such P-MOF to the hydroformylation of 1-hexene showed no significant change to the previous results obtained by other UMCM-1 derivatives giving 67% branched selectivity at 22% conversion (Supplementary Table 8), while adsorbing 70% Co, three times the molar amount of phosphino groups in the MOF. In this case, the catalyst that gives high selectivity is mostly not bound to the MOF. The evidence coming

from simulation and experiments with P-MOFs strongly suggests that the cause of the branched selectivity is not likely coming from a coordinative interaction between the Co catalyst and the MOF materials. This is also intuitively supported by the mechanism of Co-catalysed hydroformylation (Fig. 1b). Since the formation of the linear aldehyde is kinetically driven, any coordinative interaction between the catalyst and the support increases the steric hindrance around the cobalt (Fig. 1, steps **D** and **E**) and favours the formation of the linear aldehyde.

**Understanding how adsorption affects selectivity.** The application of MOFs closest to industrial scale is gas storage. The reason why MOFs are so successful in storing gaseous molecules within their pores is because the surface interaction between the material and the gas makes the packing between the molecules more efficient in the MOF micropores than in the gas phase[41]. This principle can be applied to catalysis as well. In fact, many groups have claimed that adsorption effects can play a role in the enhanced activity of catalytic reactions within the pores of MOFs due to confinement[42], a phenomenon that is known in zeolite catalysis[43,44]. To study the affinity of the reactants and products with the different MOFs, we set up a series of Monte Carlo simulations where the homogeneous liquid solution is compared to the mixture inside the pores of the crystal.

For each simulation, two periodic boxes, a cubic empty one and another reproducing the bulk crystal, were saturated with 1-hexene molecules at 30 bar and 100 °C. The number of 1-hexene molecules inside the pores of the frameworks was computed using grand canonical Monte Carlo simulations (Supplementary Table 16). Reactants and products (CO, linear heptanal and **1**) were added at infinite solution in the solvent, as one molecule per box[45–50]. During the simulation, all molecules were allowed to move according to the detailed balance at the imposed temperature, but also to swap between the two simulation boxes: the pore volume of the crystal and the homogeneous system (Supplementary Information)[51]. We could measure the affinity of the reactants and product with the MOF: an average occupancy higher than 50%, for a reactive component confined in the framework's pores shows that it is more stable in the pores than in the homogeneous phase. The smaller this probability, the more stable the component will be in the homogeneous solvent. Table 2 shows the probability related to each MOF and component and the ratio of the 1-hexene density inside the pores[52] and in the homogeneous phase.

The MOF confinement can affect the reaction in three ways: (1) an increase in the solvent density inside the pores, (2) a higher affinity with the products compared to the homogeneous phase (Supplementary Tabless 17 and 18) and (3) a lower affinity with the gas reactants. Our calculations suggest that the first two effects are due to the stronger interactions of the solvent and the aldehydes with the framework, while the third observation is caused by the formation of less interstices in the confined MOF phase, which cannot be filled with small gas molecules. One can also note that both the linear and branched products have a similar affinity with the framework in all MOFs. This evidence excludes that the branched selectivity is due to a relative stabilization of the different products in the pores, as observed in similar systems[53].

The Monte Carlo simulations point out that the concentration of the reactants within the MOF micropores is different than that in the homogeneous phase. We can achieve higher 1-hexene concentration within the pores of the MOF than under neat 1-hexene homogeneous conditions, which is remarkable. This is in line with gas storage findings and allows to access reaction conditions that are not usually achievable in homogeneous catalysis. Since the MOFs that increase branched selectivity adsorb most of the Co complex (see above), it is safe to assume

**Table 2 Affinity of the different species with the frameworks is reported as percentage occupancy (%occup.).[a]**

| Entry | MOF | 1-hexene Rel. density | H$_2$ %occup. | CO %occup. |
|---|---|---|---|---|
| 1 | UMCM-1 | 1.04 ± 0.01 | 40.0 ± 0.5% | 41.3 ± 0.5% |
| 2 | UMCM-1-NH$_2$ | 1.04 ± 0.01 | 39.1 ± 0.1% | 40.8 ± 0.5% |
| 3 | MOF-74(Zn) | 1.14 ± 0.01 | 22.4 ± 0.7% | 21.9 ± 1.4% |

[a]% occup is related to the average number of molecules of that species in the MOF's simulation box. The error is computed as standard deviation over ten independent simulations. The first column reports the relative density of 1-hexene (Rel. density) computed in the pore volume with respect to the density observed in the homogeneous simulation box (see also Supplementary Tables 17 and 18).

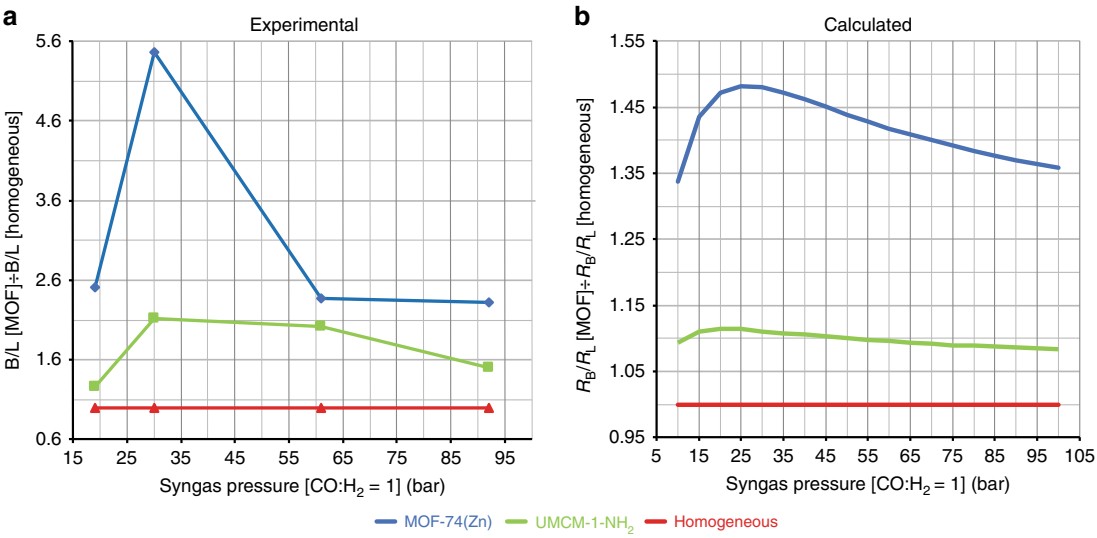

**Fig. 3 Experimental and calculated selectivity comparison. a** Experimental branched to linear ratios (B/L) with MixUMCM-1-NH$_2$ (28%) and MOF-74 (Zn) relative to the homogeneous B/L as function of syngas pressure (Supplementary Table 9). **b** Calculated relative rates of formation of the branched and linear aldehydes ($R_B/R_L$) in the MOFs UMCM-1-NH$_2$ and MOF-74(Zn) and homogeneous phase referenced to the homogeneous system as a function of syngas pressure.

that selectivity is determined within the micropores. We qualitatively identified the effect of such modified concentrations in the branched-selective hydroformylation by using published kinetic laws for the rate of formation of the branched ($R_B$) and of the linear ($R_L$) aldehyde that have been empirically determined for the hydroformylation of propene[30]. The two kinetic laws are shown in Eqs. (1) and (2) and depict the rate of formation of the branched and linear product, respectively, in the hydroformylation of propene with $k_B$ (110 °C) = 2.12 × 10$^{-7}$ (m$^3$/mol)/1.94 s, $K_{BCO}$ (110 °C) = 1.35 × 10$^{-3}$ m$^3$/mol, $k_L$ (110 °C) = 2.01 × 10$^{-7}$ (m$^3$/mol)/2.17 s and $K_{LCO}$ (110 °C) = 8.014 × 10$^{-3}$ m$^3$/mol.

$$R_B = \frac{k_B[H_2]^{0.32}[CO][Co_2(CO)_8]^{0.62}[alkene]}{(1 + K_{BCO}[CO])^2}, \quad (1)$$

$$R_L = \frac{k_L[H_2]^{0.55}[CO][Co_2(CO)_8]^{0.75}[alkene]^{0.87}}{(1 + K_{LCO}[CO])^2}. \quad (2)$$

A qualitative assessment of the effect of concentration within the pores of the MOF can be done by calculating $R_B/R_L$ rates relative to that calculated for standard homogeneous conditions, assuming that the order on the different reactants and catalysts is the same using 1-hexene and propene as olefins under the same conditions, which is a good approximation since the overall rate is not dependent on the size of the linear terminal olefin[54]. The concentration of H$_2$ and CO at different pressures in 1-hexene was calculated using the Soave modifications of the Redlich–Kwong equation (SRK) (Supplementary Table 19)[55]. We used the Monte Carlo simulation data presented above to calculate the concentrations of the reactants within the MOF

pores by multiplying the concentrations found in the homogeneous reaction by a factor Z derived in the Monte Carlo simulations (Supplementary Tables 20–23). Figure 3b shows the syngas pressure dependence of $R_B/R_L$ in UMCM-1-NH$_2$ (green) and MOF-74(Zn) (blue) compared to the homogeneous reaction (red) and depicts how we can increase $R_B/R_L$ at 15–25 bar within the pores of UMCM-1-NH$_2$ and MOF-74(Zn), the latter being superior as supported by experimental results. The local concentration of the different species and the—in part favoured —isomerization of the alkene influence the rate of formation of the branched and of the linear aldehydes. The MOF micropores create the right conditions to tune the concentration of the reactants and push the kinetic limit of the homogeneous reaction favouring the formation of the branched aldehydes. The correlation between experimental B/L ratio (Fig. 3a) and calculated data (Fig. 3b) as function of syngas pressure is evident. The deviations are more prominent in the MOF-74(Zn) case at high pressure and are caused by a lower Co uptake (36% at 61 bar, Supplementary Table 4) and by higher conversion.

In summary, MOFs can provide the right microporous environment to enhance branched selectivity by increasing the branched rate of formation while decreasing that of the linear because of concentration variations within their micropores. Not all microporous materials can push such limits since the material should adsorb the Co complex while minimizing the coordinative interaction with the catalyst and be inert towards syngas.

**Substrate scope.** Substrate scope was performed with screening aimed at achieving full conversion to show that this easy

**Table 3 Substrate scope of the Co-catalysed hydroformylation of olefins without directing groups with MOF additives and comparison with the homogeneously catalysed reaction[a,b].**

| Entry | Olefin | MixUMCM-1-NH₂ (28%)[a,c] B/L\|Conv. (%) [Oxo yield (%)] | MOF-74(Zn)[a,d] B/L\|Conv. (%) [Oxo yield (%)] | No MOF[a] B/L\|Conv. (%) [Oxo yield (%)] |
|---|---|---|---|---|
| 1 | [e] | 83/17\|75 [60] | 90/10\|85 [70] | 61/39\|99 [95] |
| 2 | [f] | 79/21\|62 [55] | 89/11\|86 [75] | 52/48\|97 [95] |
| 3 | [g] | 84/16\|84 [80] | 86/14\|81 [75] | 54/46\|97 [95] |
| 4 | [h] | 77/23\|80 [75] | 83/17\|71 [65] | 61/38\|>99 [95] |
| 5 | Ph [i] | 70/30\|15 [10][j] | 81/19\|49 [40][k] | 60/40\|58 [50][l] |

[a]Conv. = olefins conversion; Oxo yield = yield of oxo products. B/L and conversion were calculated using GC-FID with *p*-cymene as external standard. Oxo products yield was calculated by combining the mass of the raw product after reaction and the purity determined by GC-FID (see Supplementary Information). The oxo products were identified as aldehydes (Supplementary Table 10) and aldol condensation products.
[b]$Co_2(CO)_8$ (1.5 mol%) were dissolved in olefin (500 μL) and the MOF was added. The mixture was brought to 30 bar and then heated to 100 °C for 17 h.
[c]$mol_{MOF}/mol_{Co} = 0.4$.
[d]$mol_{MOF}/mol_{Co} = 3.3$.
[e]**1**:**2**:**3** ratio (homogeneous and MixUMCM-1-NH₂) = 3:1:0. **1**:**2** ratio (MOF-74(Zn)) = 2:1:0.
[f]**1**:**2**:**3** ratio = 7:2:1.
[g]**1**:**2**:**3** ratio = 7:2:1.
[h]**1**:**2**:**3** ratio = 6:2:2.
[i]**1**:**2**:**3** ratio = 7:1.5:1.5.
[j]Eleven per cent of hydrogenated olefin was detected by GC-FID and GC-MS.
[k]Ten per cent of hydrogenated olefin was detected by GC-FID and GC-MS.
[l]Five per cent of hydrogenated olefin was detected by GC-FID and GC-MS.

procedure can be applied to a range of non-functionalized branched aldehydes. Linear olefins with no directing groups from 1-hexene to 1-nonene and but-3-en-1-ylbenzene underwent hydroformylation at 100 °C, 35 bar under neat conditions and $Co_2CO_8$ (1.5 mol%) with high conversions and up to 90% branched selectivity in 17 h (Table 3). The comparison of the results in terms of selectivity with the homogeneous system is staggering. We observed in all cases an increase in branched selectivity—often between 30 and 40% increase—by simply adding a MOF to the reaction mixture showing that this protocol is flexible starting from olefins with no directing groups.

**Using microporous materials to achieve high selectivity.** By combining experiments, classical, quantum and kinetic simulations, the research shown here demonstrates the importance of micropores to push the kinetic limits and to drive the Co-catalysed hydroformylation of olefins without directing groups to exceptional branched selectivity with good substrate scope. The micropores and the chemical flexibility of MOFs create a unique combination that can be exploited to tuning the concentration of the species within the pores, which basically act as microporous reactors. The easy reaction protocol, which consists in simply adding a MOF to a homogeneously catalysed reaction, should not be overlooked.

One of the most important consequence of this work is that the methodology can be used to predict the effect of microporous co-catalysts to increase selectivity in any homogeneous or heterogeneous catalytic reaction. The requirement is that the kinetic data for the different products of the reaction is known and that the order in (at least) one of the reactants is not the same for different products. One can choose the microporous material that has the best chances of increasing selectivity (a) by appropriately selecting the ones that can adsorb the catalyst while being inert under reaction conditions and (b) by using simulations to determine how the microporous materials can change the local concentration of the selectivity-determining reactant(s) within the micropores. It is

therefore an extremely powerful tool for the design of selective catalytic heterogeneous processes in the fine chemical industry.

## Methods

**Synthesis of MOFs.** Detailed experimental methods can be found in the Supplementary Information. MixUMCM-1-NH₂ (28%) was synthesized according to a published procedure[10]. MOF-74(Zn) was synthesized according to the following procedure: in a 20 mL microwave tube, 2,5-dihydroxyterephthalic acid (200 mg; 1.01 mmol) and Zn(acac)₂·H₂O (568 mg; 2.02 mmol) were dissolved in dimethylformamide (DMF) (19 mL) and H₂O (1 mL) to give a yellow solution. The reaction mixture was stirred at 130 °C for 60 min in a Biotage Initiator+ microwave oven. The solid of the reaction mixture was filtered by membrane filter, washed with DMF, H₂O and EtOH and dried in a vacuum oven. Yield: 388 mg (81%).

**General hydroformylation procedure.** Detailed experimental methods can be found in the Supplementary Information. $Co_2(CO)_8$ (1.5 mol%) was dissolved in the olefin (500 μL), and the solution was added to the MOF in a 1 mL crimp vial. The vial was placed into a 50 mL Premex® autoclave and purged with Ar several times. Syngas pressure (CO:H₂ 1:1, 30 bar) was applied and the autoclaves heated at 100 °C for 17 h. The autoclave was allowed to cool down to room temperature before the pressure was released slowly over 15 min. The autoclave was flushed with nitrogen before it was opened to remove additional syngas for safety reasons. B/L and conversion were calculated using gas chromatography with flame ionization detector (GC-FID) with *p*-cymene as external standard. Oxo products yield was calculated by combining the mass of the raw product after reaction and the purity determined by GC-FID.

**Molecular simulations.** Detailed procedures for molecular simulations can be found in the Supplementary Information. DFT calculations were performed using the CP2K package[56] and adopting the PBEsol functional[57]. Partial charges for the classical simulation have been computed using the REPEAT scheme[58]. The RASPA software[59] was used for Monte Carlo simulations, employing the DREIDING force field[45] (extended with UFF [Universal force field] parameters[47] for Mg, Co, Ni and Mg) as dispersion parameters of the MOFs' atoms. TraPPE force field[48,49] was used to model the molecules, fitting from ab initio the missing parameters for the branched aldehydes. The pore volume of the frameworks has been computed using the software Zeo++[60].

**Kinetic analysis.** The kinetic analysis was based on the empirical rate of formations of the branched and the linear aldehydes reported elsewhere[30]. The concentrations of 1-hexene, CO and H₂ within the pores of the MOFs were calculated

by multiplying the concentration in the homogeneous phase by a factor $Z$ derived from the Monte Carlo simulations (Supplementary Table 20).

## Data availability

File inputs to reproduce the calculations of this study are available in Materials Cloud via https://doi.org/10.24435/materialscloud:2020.0007/v1. Access to any of other data can be requested on reasonable request to the corresponding author.

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

## Acknowledgements

This work was supported by a grant from the Swiss National Supercomputing Centre (CSCS) under Project no. s765. The research of D.O. was supported by the European Research Council (ERC) under the European Union's Horizon 2020 research and innovation programme (grant agreement no. 666983, MaGic). D.T. and G.B. acknowledge the National Centre of Competence in Research (NCCR) "Materials' Revolution: Computational Design and Discovery of Novel Materials (MARVEL)" of the Swiss National Science Foundation (SNSF), which funded their work at EPFL and PSI. We thank Prof. van Bokhoven for multiple discussions. We thank the Energy System Integration (ESI) platform for financial support. A special thanks to Julia Wittke for helping to calculate the solubilities of CO and $H_2$ in 1-hexene. We thank Dr. Przemyslaw Rzepka for measuring scanning electron microscopy of MOF-74(Zn) samples.

## Author contributions

The manuscript is written through contributions of all authors. All authors have given approval to the final version of the manuscript. G.B., P.G. and T.R. were supervised by M.R. G.B. and M.R. conceived the idea. G.B., P.G. and T.R. synthesized the MOFs and performed catalytic experiments. G.B. and P.G. developed the analytics of catalytic performance. M.T. developed the method to measure the Co content by ICP-MS. Molecular simulations were coordinated by B.S. D.T. and G.P. performed DFT simulations. D.O. carried out Monte Carlo simulations. M.R. carried out kinetic analysis.

## Competing interests

The authors declare no competing interests.
