## [Peer Review File · Nature Communications]

Editorial Note: Parts of this Peer Review File have been redacted as indicated as we could not obtain permission to publish the reports of reviewer 2.

Reviewers' comments:

Reviewer #1 (Remarks to the Author):

This manuscript reported that the micropores of MOFs were key factors for catalyzing hydroformylation of olefins without directing groups to branched products. When $\text{Co}_2(\text{CO})_8$ was added into the UMCM-1 or MOF-74(Zn), the catalysts could allow higher branched selectivity compared with pure homogeneous system. The authors further revealed that the micropores of MOFs with UMCM-1 and MOF-74 topologies selectively adsorbed the olefins while partially prevented adsorption of syngas by density functional theory simulations combined with kinetic models. However, based on the following questions, this manuscript is not suitable for publication in Nature communications:

1. Some obvious mistakes are found in manuscript, such as the picture of XRD of MOF-74(Ni). The “and on soluble” in the introduction part should be insoluble.
2. Although the authors emphasized the importance of pores of MOFs, but the distribution of pore size was not analyzed at all. Especially when $\text{Co}_2(\text{CO})_8$ was added into the MOFs, the detailed data about BET and pore environment was very important.
3. When pure $\text{Co}_2(\text{CO})_8$ was used to catalyze hydroformylation of olefins, they also reached around 40% conversion and 50% selectivity (Table 1). It was hard to understand why they showed nearly no catalytic activity when MIL-101(Al, Cr) was added.
4. There was no experimental data about the electronic interaction between MOFs and $\text{Co}_2(\text{CO})_8$, and XPS or EXAFS data must be performed to address this question.

In short, too many questions are not clearly answered. Although the authors spent a lot of time on theoretical calculations, no solid experimental evidences were found in the manuscript.

[REDACTED]

[REDACTED]

[REDACTED]

[REDACTED]

[REDACTED]

[REDACTED]

[REDACTED]

[REDACTED]
[REDACTED]
[REDACTED]
[REDACTED]

[REDACTED]
[REDACTED]
[REDACTED]
[REDACTED]
[REDACTED]
[REDACTED]
[REDACTED]

[REDACTED]
[REDACTED]
[REDACTED]
[REDACTED]
[REDACTED]
[REDACTED]

RESPONSE TO THE REVIEWERS

Reviewer #1 (Remarks to the Author):

This manuscript reported that the micropores of MOFs were key factors for catalyzing hydroformylation of olefins without directing groups to branched products. When $\text{Co}_2(\text{CO})_8$ was added into the UMCM-1 or MOF-74(Zn), the catalysts could allow higher branched selectivity compared with pure homogeneous system. The authors further revealed that the micropores of MOFs with UMCM-1 and MOF-74 topologies selectively adsorbed the olefins while partially prevented adsorption of syngas by density functional theory simulations combined with kinetic models. However, based on the following questions, this manuscript is not suitable for publication in Nature communications:

1. Some obvious mistakes are found in manuscript, such as the picture of XRD of MOF-74(Ni). The “and on soluble” in the introduction part should be insoluble.

1. Typos have been corrected. The picture of XRD was changed with a high resolution one and corrected the axis title.

2. Although the authors emphasized the importance of pores of MOFs, but the distribution of pore size was not analyzed at all. Especially when $\text{Co}_2(\text{CO})_8$ was added into the MOFs, the detailed data about BET and pore environment was very important.

2. As requested by the reviewer, we performed nitrogen physisorption experiments before and after catalysis. The BET number of MixUMCM-1-NH₂ (28%) decreased from 1756 m²/g to 600 m²/g after catalysis, whereas the one of MOF-74(Zn) decreased from 930 m²/g to 30 m²/g. The Powder X-ray diffraction patterns before and after catalysis showed that the frameworks were intact in both cases. The lower surface areas are the results of catalyst adsorption combined with residual organic compounds blocking the pores of the MOFs (see also point 12 below). We added a section with such data in the supplementary materials (Table S1) and added the following sentence to the manuscript.

“Both MixUMCM-1-NH₂ (28%) and MOF-74(Zn) retained crystallinity after catalysis as shown by powder X-ray diffraction. The BET number of MixUMCM-1-NH₂ (28%) decreased from 1760 m²/g to 600 m²/g after catalysis, whereas the one of MOF-74(Zn) decreased from 990 m²/g to 20 m²/g: This is caused by the adsorption of the catalyst and of organic products such as aldol compounds in the small pores (Supplementary Notes).”

3. When pure $\text{Co}_2(\text{CO})_8$ was used to catalyze hydroformylation of olefins, they also reached around 40% conversion and 50% selectivity (Table 1). It was hard to understand why they showed nearly no catalytic activity when MIL-101(Al, Cr) was added.

3. We do not have an explanation as to why MIL-101 topologies have a detrimental effect to catalysis. We suspect that an electronic interaction between the inorganic nodes of MIL-101 and the catalyst that kills catalytic activity as can be shown by the following publications.

Sakthivel, A., Mahato, N. R., Baskaran, T., & Christopher, J.

<https://doi.org/10.1016/j.catcom.2015.02.024>

Tucci, E. R. (1985). <http://dx.doi.org/10.1021/i300017a008>

A full explanation is out of the scope for this manuscript, which is already dense with concepts. We added the following sentence to the manuscript.

“The inactivity of MIL-101 and the zeolite might be explained by an electronic interaction between the metal open sites in the MOF and the Co catalyst. The detrimental effect of Al/amines and of Cr to the activity in Co-catalysed hydroformylation is described in the literature.^{32,33}”

4. There was no experimental data about the electronic interaction between MOFs and $\text{Co}_2(\text{CO})_8$, and XPS or EXAFS data must be performed to address this question.

4. Experimental data on the electronic interaction would be insightful. However, such experiments are not trivial at all. Ex-situ experiments (neither XPS or XAS) would not provide any useful information since in hydroformylation, as soon as the catalyst is out of the syngas atmosphere, the chemical speciation of the catalyst becomes completely different and any significance of the experiment is lost. For a reference: Nelsen, E. R.; Landis, C. R., Interception and Characterization of Alkyl and Acyl Complexes in Rhodium-Catalyzed Hydroformylation of Styrene. J. Am. Chem. Soc. 2013, 135 (26), 9636-9639. The only option would be to perform in-situ experiments, which are experimentally challenging and require an in-situ 3 phases (liquid gas solid) investigation which needs long beamtime and is out of the scope for this manuscript.

In short, too many questions are not clearly answered. Although the authors spent a lot of time on theoretical calculations, no solid experimental evidences were found in the manuscript.

5. We disagree with the reviewer on this point. The manuscript's experimental evidence comes from catalytic results, which are in a good agreement with adsorption and kinetic models. To make this point more clear, we changed Figure 3 and compared kinetic analysis with experimental data. We also added the following sentences to the manuscript:

"The correlation between experimental B/L ratio (Fig. 3a) and calculated data (Fig.- 3b) as function of syngas pressure is evident. The deviations are more prominent in the MOF-74(Zn) case at high pressure and are caused by a lower Co uptake (36% at 61 bar) and by higher conversion. "

Fig. 3 a) Experimental branched to linear ratios (b/l) with UMCM-1-NH₂ and MOF-74(Zn) relative to the homogeneous b/l as function of syngas pressure (Supplementary Table 6). b) Calculated relative rates of formation of the branched and linear aldehydes (R_B/R_L) in the MOFs UMCM-1-NH₂ and MOF-74(Zn) and homogeneous phase referenced to the homogeneous system as function of syngas pressure.

[REDACTED]

[REDACTED]

[Redacted text block]

[Redacted text block]

[Redacted text block]

[Redacted text block]

[Redacted text block]

[Redacted text block]

[Redacted]

[Redacted]

[Redacted]

[Redacted]

[Redacted text block]

[Redacted text block]

[Redacted text block]

[Redacted text block]

[Redacted text block]

[Redacted text block]

Reviewers' comments:

Reviewer #1 (Remarks to the Author):

The authors responded the questions raised by the reviewers in detail. However, based on the following points or problems, this manuscript is unsuitable for publication in Nature communications:

1. Some obvious mistakes could be found, for example, the angle does not exceed 40° in all XRD data, but the authors wrote "a 2θ range of $4-70^\circ$ " in physical methods part. The extra MixUMCM-1-NH₂(28%) in the powder X-ray diffractograms of MixUMCM-1-NH₂(28%) must be deleted. The " $k_L(110^\circ\text{C}) = 2.01 \cdot 10^{-7}(\text{m}^3 \cdot \text{mol}^{-1}) \cdot 2.17 \text{s}^{-1}$ " in kinetic analysis part should be $k_L(110^\circ\text{C}) = 2.01 \cdot 10^{-7}(\text{m}^3 \cdot \text{mol}^{-1}) \cdot 2.17 \text{s}^{-1}$.
2. The authors just provided data of specific surface area, but they continued to ignore presentation of the pore size distribution. Neither BET curves nor the distribution of pore size was added, and these data are very important to understand the relationship between structure and catalytic activity.
3. The authors emphasized that MOFs could be used as kinetic modulators for branched selectivity in hydroformylation. Most importantly, these materials must be characterized clearly, XRD data of synthesized MOFs is not enough, and the morphology and structure information of MOFs need to be added.
4. In Table S1, The authors mentioned that BET value decreased sharply after catalysis and the lower surface areas were the results of catalyst adsorption combined with residual organic compounds blocking the pores of the MOFs. So nearly no pore existed in MOFs after reaction. There is no recycling ability for this type of heterogeneous catalysts. Another serious problem was that when Co₂(CO)₈ was loaded into the pores of MOFs in catalytic process, the substrates with large size could not enter to react with catalytic centers, and great attention should be paid to the pore size distribution of the mentioned MOFs.
5. This manuscript mentioned that general procedure for hydroformylation of 1-hexene was as follows: A stock solution of Co₂(CO)₈ in 1-hexene was prepared inside the glove box. In a 1.5 mL GC grimp vial the MOF was weight in and activated at 150°C overnight. The MOF was then suspended in 250 μL Co/hexene stock solution and the vial was closed. But during the reaction process, two states of catalytic centers were present in solution, one was free Co₂(CO)₈, and another one was Co₂(CO)₈ enwrapped in the pores of MOFs, It is difficult to draw conclusion that the micropores of MOFs with UMCM-1 and MOF-74 topologies selectively adsorb the olefins while partially preventing the adsorption of syngas leading to high branched selectivity, even though the authors thought correlation between experimental B/L ratio (Fig. 3a) and calculated data (Fig.- 3b) as function of syngas pressure was good. In my opinion,

the better way is that $\text{Co}_2(\text{CO})_8$ was mixed with MOFs and free $\text{Co}_2(\text{CO})_8$ was washed off. I suggest that the authors test the catalytic performance of $\text{Co}_2(\text{CO})_8/\text{MOFs}$ to show if they could be used as kinetic modulators.

6. The manuscript showed that nearly no catalytic activity when MIL-101(Al, Cr) was added. But not any characterization data for MIL-101(Al, Cr) was present at all. It is still unknown if MIL-101(Al, Cr) could be synthesized successfully.

7. There was still no experimental data about the electronic interaction between MOFs and $\text{Co}_2(\text{CO})_8$. The XPS data of $\text{Co}_2(\text{CO})_8$ and $\text{Co}_2(\text{CO})_8/\text{MOFs}$ could show if there is electronic interaction between them rather than simply providing calculation data.

8. Fuzzy experimental conditions were present in the manuscript, such as “Inside the glove box, $\text{Co}_2(\text{CO})_8$ (18 mg, 108 μmol) and MixUMCM-1-PPh₂ (29 mol%) were weight into a 10 mL grimp vial and toluene (5.0 mL) was added.” No exact amount of MixUMCM-1-PPh₂ was present. And the data of GC-MS data must be provided in the supporting information to show catalytic results accurately.

[REDACTED]

[REDACTED]

[REDACTED]

[REDACTED]

[REDACTED]

[REDACTED]

[REDACTED]

[REDACTED]

[REDACTED]

[REDACTED]

RESPONSE TO THE REVIEWERS

We thank the reviewers for the comments that have improved the quality of the manuscript. We extensively revised the manuscript and addressed the reviewers' comments. Authors' response is provided in bold. The additions and modifications of the main text are highlighted in green in the main text, supplementary material, and below.

Reviewer #1 (Remarks to the Author):

The authors responded the questions raised by the reviewers in detail. However, based on the following points or problems, this manuscript is unsuitable for publication in Nature communications:

1. Some obvious mistakes could be found, for example, the angle does not exceed 40° in all XRD data, but the authors wrote "a 2θ range of 4-70°" in physical methods part. The extra MixUMCM-1-NH₂(28%) in the powder X-ray diffractograms of MixUMCM-1-NH₂(28%) must be deleted. The " $k_L(110^\circ\text{C}) = 2.01 \cdot 10^{-7}(\text{m}^3 \cdot \text{mol}^{-1}) \cdot 2.17\text{s}^{-1}$ " in kinetic analysis part should be $k_L(110^\circ\text{C}) = 2.01 \cdot 10^{-7}(\text{m}^3 \cdot \text{mol}^{-1}) \cdot 2.17\text{s}^{-1}$.

Such typos have been thoroughly checked and modified.

2. The authors just provided data of specific surface area, but they continued to ignore presentation of the pore size distribution. Neither BET curves nor the distribution of pore size was added, and these data are very important to understand the relationship between structure and catalytic activity.

We now added pore size distribution of the MOFs before and after catalysis in the Supplementary Notes. We re-synthesized MixUMCM-1-NH₂ (28%) and obtained a higher surface area material. Catalytic results were identical to those of Table 1 run 5 within a +/- 1% difference so we decided to leave the result in Table 1 unvaried. We modified a sentence in the main text:

The BET number of MixUMCM-1-NH₂ (28%) changed from 2870 m²/g to 2980 m²/g after catalysis, whereas the one of MOF-74(Zn) decreased from 1000 m²/g to 150 m²/g: The lower surface area in MOF-74 after catalysis is caused by the adsorption of the catalyst and of organic products such as aldol compounds (Supplementary Notes), which could not be removed by extensive washing, as also evidenced by pore size distributions before and after catalysis and by the recycling experiments below.

3. The authors emphasized that MOFs could be used as kinetic modulators for branched selectivity in hydroformylation. Most importantly, these materials must be characterized clearly, XRD data of synthesized MOFs is not enough, and the morphology and structure information of MOFs need to be added.

We provided information on the crystallite size and shape by measuring optical microscopy and SEM. These analyses are presented in the supplementary material.

Figure S 1 Optical microscope image of MixUMCM-1-NH₂ (28%) (left) and SEM image of MOF-74(Zn) (right). Both MOFs show a needle shape. The crystal size is between 1.5 and 4.3 μm for MOF-74(Zn) and around 390-680 μm for MixUMCM-1-NH₂ (28%).

4. In Table S1, The authors mentioned that BET value decreased sharply after catalysis and the lower surface areas were the results of catalyst adsorption combined with residual organic compounds blocking the pores of the MOFs. So nearly no pore existed in MOFs after reaction. There is no recycling ability for this type of heterogeneous catalysts. Another serious problem was that when Co₂(CO)₈ was loaded into the pores of MOFs in catalytic process, the substrates with large size could not enter to react with catalytic centers, and great attention should be paid to the pore size distribution of the mentioned MOFs.

Reviewer 2 also asked for recycling data. We provided recycling experiments of both impregnated catalysts (See Point 5 below) and of the MOFs after extensive washing. The results show that the cobalt adsorbed in the MOFs after catalysis is not active, as expected given the necessity of a syngas atmosphere for a stable and active catalytic system as discussed in the previous revision. Most probably the best reactor system for such reaction is a continuous flow reactor, the development of which is out of the scope for this manuscript. The extensively washed MOFs after recycling had lower branched selectivity than the fresh MOF, but higher than the homogeneous reaction. These data are useful additional information and do not take anything conceptual out of the manuscript as also stated by reviewer 2 below. We added the following paragraph to the manuscript:

After the first catalytic runs in Table 1 Entries 5 and 6, we extensively washed the MOFs and tested their recyclability for a second catalytic run. MixUMCM-1-NH₂ (28%) showed 5% conversion and 64% branched selectivity, whereas MOF-74(Zn) exhibited 68% selectivity at 35% conversion (Supplementary Table 9). Even though branched selectivity decreased in both recycling experiments compared to the first run, it was still higher than in the homogeneous reaction. Higher conversion could be obtained with MOF-74(Zn). All recycling results suggest that there are trace compounds that are hard to wash out and may modify the adsorption properties of the MOFs causing lower selectivity as suggested also by BET after catalysis (see above). The recycling of impregnated Co@MOF catalysts with minimal washing – to keep the Co inside the pores – resulted in inactive catalysts (Supplementary Table 8).

5. This manuscript mentioned that general procedure for hydroformylation of 1-hexene was as follows: A stock solution of Co₂(CO)₈ in 1-hexene was prepared inside the glove box. In a 1.5 mL GC crimp vial the MOF was weight in and activated at 150 °C overnight. The MOF was then suspended in 250 μL Co/hexene stock solution and the vial was closed. But during the reaction

process, two states of catalytic centers were present in solution, one was free $\text{Co}_2(\text{CO})_8$, and another one was $\text{Co}_2(\text{CO})_8$ enwrapped in the pores of MOFs, It is difficult to draw conclusion that the micropores of MOFs with UMCM-1 and MOF-74 topologies selectively adsorb the olefins while partially preventing the adsorption of syngas leading to high branched selectivity, even though the authors thought correlation between experimental B/L ratio (Fig. 3a) and calculated data (Fig.- 3b) as function of syngas pressure was good. In my opinion, the better way is that $\text{Co}_2(\text{CO})_8$ was mixed with MOFs and free $\text{Co}_2(\text{CO})_8$ was washed off. I suggest that the authors test the catalytic performance of $\text{Co}_2(\text{CO})_8/\text{MOFs}$ to show if they could be used as kinetic modulators.

We thank the reviewer for the insightful comment. Washing out the MOFs after treatment with $\text{Co}_2(\text{CO})_8$ desorbed all pre-catalyst from the MOF. We therefore prepared several cobalt-loaded MOFs through incipient wetness impregnation with solutions of $\text{Co}_2(\text{CO})_8$ at different concentrations. After adding 1-hexene for catalysis, the Co was mostly in solution. We attempted catalysis but such protocol gave poorer performance in terms of selectivity as shown in Supplementary Table 7. We used the old protocol for substrate scope. We added a paragraph in the manuscript:

We attempted incipient wetness impregnation of the $\text{Co}_2(\text{CO})_8$ pre-catalyst in dichloromethane at MOF/Co molar ratio of 0.7 and 0.3 with UMCM-1- NH_2 and MOF-74(Zn), respectively, to check whether the pre-adsorbed metal would lead to higher selectivity. Once 1-hexene was added to the impregnated MOF, a strong coloration of the homogeneous solution to dark brown was observed in all experiments indicating that the Co complex was preferentially in solution. Catalytic results under such conditions showed up to 41% conversion and 68% branched selectivity 0.46 mol_{Co}% for UMCM-1- NH_2 and up to 63% conversion and 68% branched selectivity with a 0.6 mol_{Co}% loading in the case of MOF-74(Zn) (Supplementary Table 7), showing that higher selectivity can be obtained by adding the MOF to the reaction mixture rather than with pre-formed impregnated pre-catalyst. This is attributed to the relatively low chemical stability of $\text{Co}_2(\text{CO})_8$ ³³ and suggests that it is an intermediate of the catalytic cycle that preferentially adsorbs within MOFs rather than the pre-catalyst.

6. The manuscript showed that nearly no catalytic activity when MIL-101(Al, Cr) was added. But not any characterization data for MIL-101(Al, Cr) was present at all. It is still unknown if MIL-101(Al, Cr) could be synthesized successfully.

We added such characterization data in the Supplementary Material.

7. There was still no experimental data about the electronic interaction between MOFs and $\text{Co}_2(\text{CO})_8$. The XPS data of $\text{Co}_2(\text{CO})_8$ and $\text{Co}_2(\text{CO})_8/\text{MOFs}$ could show if there is electronic interaction between them rather than simply providing calculation data.

As discussed in the first reply, any meaningful data on the interaction between MOF and the catalyst should be performed *in-situ*, which is out of the scope for this manuscript. We talked to colleagues that are experts in XPS and such measurement will not lead to any meaningful data on the interaction between MOF and catalyst due the instability of the cobalt precursor and its sublimation in vacuum. We provided fourier transform infrared spectroscopy (FT-IR) of UMCM-1- NH_2 impregnated with $\text{Co}_2(\text{CO})_8$ showed CO stretches between 1800 cm^{-1} and 2000 cm^{-1} indicating little – if any – electronic interaction between the pre-catalyst and the MOF. FT-IR of MOF-74 impregnated with $\text{Co}_2(\text{CO})_8$ showed no CO stretches between 1800 cm^{-1} and 2000 cm^{-1} with concomitant colour change from dark black to light grey as soon as the sample was out of protective atmosphere for measurement. This indicated fast decomposition of $\text{Co}_2(\text{CO})_8$ under O_2 and/or moisture. The IR spectra and comparison with the $\text{Co}_2(\text{CO})_8$ and the MOFs are shown below.

We added FT-IR spectra and comments in the Supplementary Notes.

8. Fuzzy experimental conditions were present in the manuscript, such as “Inside the glove box, Co₂(CO)₈ (18 mg, 108 μmol) and MixUMCM-1-PPh₂ (29 mol%) were weight into a 10 mL crimp vial and toluene (5.0 mL) was added.” No exact amount of MixUMCM-1-PPh₂ was present. And the data of GC-MS data must be provided in the supporting information to show catalytic results accurately.

Such mistakes have been thoroughly checked and modified. We also added GC-MS data to show aldol products.

[Redacted text block]

[Redacted text block]

[Redacted text block]

[Redacted text block]

[Redacted text block]

[Redacted text block]

[Redacted text block]

[Redacted text block]

Reviewers' comments:

Reviewer #1 (Remarks to the Author):

After carefully reviewing the manuscript, the reviewer believe that this manuscript is not suitable for publication in Nature Communications.

1. The provided BET curves and distribution of pore sizes are abnormal, especially the data of MOF-74(Co) are something wrong. On one hand, the curve raises from 0 to above 200 sharply without any points in this part is unreasonable; on the other hand, the data of distribution of pore size about MOF-74(Co) show no microporous structures at all whereas the BET value could reach above 1000, which seems incredible. The main highlight that the authors mentioned is about tuning branched selectivity by using metal-organic frameworks as kinetic modulators, while the corresponding experimental evidences do not support this viewpoint well.

2. Except that the BET data of MOF-74 is unreasonable, it is very important to make sure that all materials synthesized in the manuscript are correct and then one can compare the catalytic performance carefully. The authors just provided the XRD data of MIL-101(Al/Cr), which only indicated that some corresponding crystals were synthesized. Many works are missed, for instance, the pores and corresponding information are not clear, and other impurity might exist in pores of MOF. The catalytic results are meaningful only after all the materials are characterized accurately.

3. The authors stated that Fourier transform infrared spectroscopy (FTIR) of UMCM-1 -NH₂ impregnated with Co₂(CO)₈ showed CO stretches between 1800 cm⁻¹ and 2000 cm⁻¹, indicating little - if any - electronic interaction between the pre-catalyst and the MOF. The statement is obviously wrong, and the data of FTIR only show information of functional groups and could not support this view.

4. Synthetic materials reported in the manuscript had poor cycle capability, which did not give enough guidelines to design new, efficient and stable heterogeneous catalysts. What is more important, no clear correlation between the structure and catalytic activity could be seen in this manuscript.

Reviewer #3 (Remarks to the Author):

I have had the opportunity to read this manuscript and, while I was not involved in the original review process, also the 'response to reviewers' document. I find that the manuscript presents some new and important science. The reviewers raised many pertinent points in their reviews. These have been thoughtfully considered by the authors, leading to an enhanced manuscript.

The main strength of the scientific work is the fact that well known homogeneous catalysts can be embedded in MOFs to produce heterogeneous catalysts. The MOFs are carefully designed for this purpose. And the catalytic outcome is altered in useful and interesting ways. It appears that the catalyst is stabilised in the MOF, however I think there is plenty of follow-up work that can be done to better establish this. Additionally, the alkene conversion may need to be raised before the method appeals to researchers in the area of alkene hydroformylation. While outside of the scope of the present manuscript, this would connect the research to desired applications in industry. The current work provides an excellent foundation for this future work in the form of a proof of principle and useful methodologies.

It appears that many important details were omitted from the original version of the manuscript, but this had been rectified to the point where other researchers could reasonably reproduce the results. While the ESI could be more attractively presented, it contains sufficient information. The amended references are appropriate.

RESPONSE TO THE REVIEWERS

We thank the reviewers for the comments. We revised the manuscript and addressed the reviewers' comments. Authors' response is provided in bold. The additions and modifications of the main text are highlighted in green in the main text and supplementary notes.

REVIEWER 1

The provided BET curves and distribution of pore sizes are abnormal, especially the data of MOF-74(Co) are something wrong. On one hand, the curve raises from 0 to above 200 sharply without any points in this part is unreasonable; on the other hand, the data of distribution of pore size about MOF-74(Co) show no microporous structures at all whereas the BET value could reach above 1000, which seems incredible. The main highlight that the authors mentioned is about tuning branched selectivity by using metal-organic frameworks as kinetic modulators, while the corresponding experimental evidences do not support this viewpoint well.

We disagree with the reviewer that our data are wrong or abnormal. We use a porosimeter that is accurate for pores above 9 Å, and as the medium pore width is 9.7 Å in MOF-74, this is the right equipment to use. Lower pressure data are not relevant for the determination of the BET number (Chem. Mater. 2017, 29, 1, 26-39). Pore size analysis of MOF-74(Zn) before catalysis clearly shows micropores below 10Å with differential dV/dw up to $0.035 \text{ cm}^3/\text{g}\cdot\text{Å}$, which is reduced to $0.005 \text{ cm}^3/\text{g}\cdot\text{Å}$ after catalysis. This is in line with BET numbers and is helpful to characterize pores before and after catalysis showing that the adsorption of the catalyst and of organic products such as aldol compounds after catalysis reduces adsorption in the micropores. Physisorption data (BET and pore size analysis) relevant for catalysis are reported for MOF-74(Zn), not MOF-74(Co), maybe the reviewer is confusing the two materials.

2. Except that the BET data of MOF-74 is unreasonable, it is very important to make sure that all materials synthesized in the manuscript are correct and then one can compare the catalytic performance carefully. The authors just provided the XRD data of MIL-101(Al/Cr), which only indicated that some corresponding crystals were synthesized. Many works are missed, for instance, the pores and corresponding information are not clear, and other impurity might exist in pores of MOF. The catalytic results are meaningful only after all the materials are characterized accurately.

We disagree on this point. All MOFs reported in this work have been characterized in line what is common in the field (Chem. Mater. 2017, 29, 1, 26-39). We feel that we may not have organized the supplementary notes sufficiently clearly (see point of reviewer 3). This is the only explanation we have why the reviewer must have missed that all MOFs in this study have been characterized by both XRD and BET as shown in the supplementary notes. We revised the supplementary notes to make them easier to read.

3. The authors stated that Fourier transform infrared spectroscopy (FTIR) of UMCM-1 -NH₂ impregnated with Co₂(CO)₈ showed CO stretches between 1800 cm⁻¹ and 2000 cm⁻¹, indicating little - if any - electronic interaction between the pre-catalyst and the MOF. The statement is obviously wrong, and the data of FTIR only show information of functional groups and could not support this view.

This is a good point, the sentence is confusing. The sentence now reads: "Fourier transform infrared spectroscopy (FT-IR) of UMCM-1-NH₂ impregnated with Co₂(CO)₈ showed CO stretches between 1800 cm⁻¹ and 2000 cm⁻¹ similar to those of the neat pre-catalyst Co₂(CO)₈.

This indicates that the structure of the pre-catalyst is similar in the presence of the MOF or without it. "

4. Synthetic materials reported in the manuscript had poor cycle capability, which did not give enough guidelines to design new, efficient and stable heterogeneous catalysts. What is more important, no clear correlation between the structure and catalytic activity could be seen in this manuscript.

We disagree with the reviewer on this point. The importance of our work is on how microporous materials can change selectivity by modulating kinetics exploiting the pore systems and how this concept can be applied to make novel selective catalysts:

- **We provided computational and experimental results that give substantial evidence that an interaction between the catalysts and the MOFs that show kinetic modulation is minimal.**
- **The structure of the catalyst is likely the same as the homogeneous catalyst, for which there is plenty of literature on its structure and mechanism (Chem. Rev. 2009, 109, 9, 4272-428).**
- **Figure 3 in the main text on simulated kinetic data and experimental branched/linear ratio shows a very good correlation between experimental and modelled data.**

All such combined data give a strong correlation between structure of the catalyst, of the MOF and catalytic activity.

Editorial note: Reviewer 2 is replacing the previous Reviewer 2 and is actually Reviewer 3.

REVIEWER 2

I have had the opportunity to read this manuscript and, while I was not involved in the original review process, also the 'response to reviewers' document. I find that the manuscript presents some new and important science. The reviewers raised many pertinent points in their reviews. These have been thoughtfully considered by the authors, leading to an enhanced manuscript.

The main strength of the scientific work is the fact that well known homogeneous catalysts can be embedded in MOFs to produce heterogeneous catalysts. The MOFs are carefully designed for this purpose. And the catalytic outcome is altered in useful and interesting ways. It appears that the catalyst is stabilised in the MOF, however I think there is plenty of follow-up work that can be done to better establish this. Additionally, the alkene conversion may need to be raised before the method appeals to researchers in the area of alkene hydroformylation. While outside of the scope of the present manuscript, this would connect the research to desired applications in industry. The current work provides an excellent foundation for this future work in the form of a proof of principle and useful methodologies.

It appears that many important details were omitted from the original version of the manuscript, but this had been rectified to the point where other researchers could reasonably reproduce the results.

While the ESI could be more attractively presented, it contains sufficient information. The amended references are appropriate.

We thank the reviewer for the positive remarks. We modified the structure of the supplementary notes to make them easier to read.